# In situ electrochemical regeneration of nanogap hotspots for continuously reusable ultrathin SERS sensors

Sarah May Sibug-Torres [1], David-Benjamin Grys[1], Gyeongwon Kang [1,2], Marika Niihori[1], Elle Wyatt[1], Nicolas Spiesshofer[1], Ashleigh Ruane [1], Bart de Nijs [1] & Jeremy J. Baumberg [1] ✉

Surface-enhanced Raman spectroscopy (SERS) harnesses the confinement of light into metallic nanoscale hotspots to achieve highly sensitive label-free molecular detection that can be applied for a broad range of sensing applications. However, challenges related to irreversible analyte binding, substrate reproducibility, fouling, and degradation hinder its widespread adoption. Here we show how in-situ electrochemical regeneration can rapidly and precisely reform the nanogap hotspots to enable the continuous reuse of gold nanoparticle monolayers for SERS. Applying an oxidising potential of +1.5 V (vs Ag/AgCl) for 10 s strips a broad range of adsorbates from the nanogaps and forms a metastable oxide layer of few-monolayer thickness. Subsequent application of a reducing potential of −0.80 V for 5 s in the presence of a nanogap-stabilising molecular scaffold, cucurbit[5]uril, reproducibly regenerates the optimal plasmonic properties with SERS enhancement factors ≈10⁶. The regeneration of the nanogap hotspots allows these SERS substrates to be reused over multiple cycles, demonstrating ≈5% relative standard deviation over at least 30 cycles of analyte detection and regeneration. Such continuous and reliable SERS-based flow analysis accesses diverse applications from environmental monitoring to medical diagnostics.

Surface-enhanced Raman Spectroscopy (SERS) has emerged as a powerful analytical technique, unlocking prospects for sensitive label-free molecular fingerprinting across various fields including forensic analysis[1], environmental monitoring[2], biomedical diagnostics[3–5], and food quality control[6]. SERS is based on the enhancement of Raman scattering that arises from strong electromagnetic fields generated by the optical excitation of localized surface plasmons ('hotspots') on nanostructured metal substrates. The SERS from analytes adsorbed at the hotspots can be enhanced by $10^3$–$10^{10}$, enabling the identification and quantification of trace concentrations of analytes[7]. However, the same sensitivity that endows SERS with its analytical capability renders it particularly susceptible to irreproducibility from variations in substrate nanoscale fabrication[8] and contamination[9,10]. Achieving continuous high-throughput SERS measurements is further compromised by irreversible analyte binding at the hotspots (the SERS 'memory effect'), resulting in the need for frequent substrate replacement[11]. This presents insuperable practical issues, including high operating costs, poor resource utilization, and waste generation. To address these challenges, a promising route is to develop effective SERS substrate recycling strategies to enable multiple sample analyses on the same platform, thus delivering reproducible high-throughput SERS-based analysis.

To reuse SERS sensors, a key requirement is repeatedly removing a broad range of analytes without altering the crucial nanoscale

[1]NanoPhotonics Centre, Cavendish Laboratory, Department of Physics, JJ Thompson Avenue, University of Cambridge, Cambridge CB3 OHE, UK. [2]Present address: Department of Chemistry, Kangwon National University, Chuncheon 24341, South Korea. ✉e-mail: jjb12@cam.ac.uk

morphology which controls the SERS hotspots and directly affects analyte signal intensities[8]. Various strategies have been proposed[12], including metal etching and redeposition[13,14]; analyte decomposition with thermal[15,16], plasma[17,18], or photochemical[19,20] treatments; use of protective polymers[21–23]; solvent rinsing[24–26]; analyte displacement with competitive binding[27–31]; and electrochemical methods[28,32–39] (see Supplementary Table 1 for comprehensive comparison). However, previous methods that remove strongly adsorbed molecules can induce significant morphological changes that degrade the SERS hotspots. Electrochemical (EC) strategies are the most promising in terms of speed, in situ implementation, and the range of analytes that can be removed, making EC strategies promising for rapid, high-throughput sequential sample analysis. Still, maintaining analyte signal reproducibility across multiple cleaning cycles remains unfeasible due to the degradation of the SERS hotspots[40]. While EC cleaning via oxidation and reduction removes adsorbates from metal surfaces[41–43], it leads to significant nanoscale surface restructuring[44,45]. Achieving reliable scalable SERS-based analysis thus necessitates advances in the precision control of nanoscale morphology to repeatedly regenerate SERS hotspots.

Here we present an in situ electrochemical SERS hotspot regeneration scheme ('ReSERS') for the continuous reuse of thin-film gold nanoparticle (AuNP) SERS substrates. Previously, we demonstrated a low-cost scalable method to prepare thin-film SERS substrates through bottom-up self-assembly of AuNPs using a rigid molecular scaffold, cucurbit[$n$]uril (CB[$n$], $n = 5-8$), which yields highly reproducible sub-1 nm nanogap hotspots across multi-layer AuNP aggregates ('MLagg')[9,46,47]. We now show that by integrating MLaggs within electrochemical SERS (EC-SERS) flow cells, the ligand-stabilized ('scaffolded') nanogap hotspots can be anodically stripped of molecular adsorbates with the simultaneous formation of a meta-stable gold oxide plug. A second step precisely regenerates the hotspots by reducing the few-monolayer oxide layer while re-stabilizing the metallic nanogaps with the CB[$n$] scaffolding molecule. Using this cleaning and regeneration protocol, the same SERS substrate can be reused with a high level of reproducibility and number of regeneration cycles. We find that the key to reproducibility is the reintroduction of the molecular scaffold, which precisely reforms the nanogap hotspots after oxidative cleaning. Efficient ReSERS enables direct quantitative analysis of complex samples without substrate fouling and allows us to identify key criteria for its application to different SERS substrates. Fully regenerating nanogaps thus gives continuous high-throughput SERS measurements, with broad applications in health, environment, industrial process, and quality monitoring.

## Results

### SERS substrate preparation and oxidative cleaning

To prepare a SERS substrate with precisely-controlled nanogap spacing, we self-assemble citrate-stabilized 80 nm AuNPs with CB[5], a barrel-shaped molecule which binds AuNPs together through its carbonyl portals[8,46] (Fig. 1a, Supplementary Fig. 1a). The AuNP:CB[5] assembly can be concentrated and deposited as a thin film, yielding close-packed multi-layer AuNP aggregates with $0.90 \pm 0.05$ nm interparticle spacings defined by the CB[5] scaffold[9,47,48]. These sub-nm hotspots give an enhancement factor of $\approx 10^6$ (see Methods). Such MLaggs also have multiple advantages for delivering in-flow SERS sensing, such as their ease of integration into flow cell systems and their capability to be probed optically from the backside when deposited on transparent solid supports. The MLaggs further allow for tuning the surface chemistry[9]. Interfering native AuNP ligands such as citrate can be completely removed and replaced with desired scaffolding molecules. One way to clean MLagg surfaces is to use oxygen plasma cleaning (O$_2$-PC), followed by a 're-scaffolding' step in which diverse molecules are used to redefine the nanogaps (Supplementary Fig. 1b)[9]. Here, ligands and contaminants are

decomposed in O$_2$-PC by bombarding the surface with reactive oxygen species, which then react with the bare Au surface to form an oxide layer. This few-monolayer thick Au oxide serves a key role in plugging and stabilizing the nanogap when no stabilizing ligands are present. The hotspots can then be regenerated by decomposing the Au oxide in the presence of a desired scaffold such as CB[5], which rebinds to and restabilises the nanogap. However, plasma treatment requires an hour to remove analytes from the hotspots (Supplementary Fig. 3) and requires a plasma source incompatible with continuous and/or low-cost measurements.

Adsorbate removal and Au oxide formation are here achieved electrochemically, offering a strategy to control the surface properties of SERS substrates in situ. By setting the SERS substrate as a working electrode in a three-electrode electrochemical cell, a potential is applied to control the surface charge and induce local reactions directly on its surface[49,50]. With a Au-based SERS substrate, the application of an anodic potential oxidizes and/or desorbs adsorbates as well as oxidises the Au. This process is rapid (seconds) even for nanogaps, as the oxidation process is driven directly at the Au surface[51], in contrast to O$_2$-PC which relies on reactive oxygen surface collisions. Electrochemical reduction of the Au oxide layer then occurs upon application of a cathodic potential. Aside from analyte removal, EC-SERS can modulate analyte binding, increasing SERS signals by up to 10-fold through the enhanced adsorption of analytes[49]. While EC-SERS is highly promising for offering rapid in situ control of both analyte binding and removal, electrochemical oxidative cleaning has not so far been attempted with nanogap re-scaffolding to regenerate SERS hotspots. We thus examine multiple analyte detection, cleaning, and regeneration cycles to assess both the stripping capability and regeneration repeatability.

### Electrochemical stability of MLagg-CB[5]

To enable electrochemical control of their surface potential, MLaggs are deposited on a fluorine-doped tin oxide (FTO)-coated glass slide and assembled in an EC-SERS flow cell as the working electrode (Fig. 1b, c, Supplementary Fig. 4). SERS spectra are recorded by illuminating the MLagg with a 785 nm laser through the transparent FTO-coated glass, facilitating in situ monitoring of the nanogap while simultaneously controlling the applied potential.

The electrochemical stability of MLagg-CB[5] hotspots is initially explored by time-resolved SERS while sweeping between oxidizing and reducing potentials using cyclic voltammetry (Supplementary Note 1). Desorption of CB[5] is observed during the anodic scan (0.2 to +1.5 V) in 50 mM potassium phosphate buffer (pH 7.0) with the formation of Au oxide from +1.0 to +1.5 V (Supplementary Fig. 5). Upon the return cathodic scan, the oxide layer is reduced with re-adsorption of some CB[5] around +0.35 V. However, repeated oxidation and reduction results in the continued desorption of CB[5] from the nanogap along with a gradual loss in SERS activity. The addition of excess CB[5] in the buffer solution better stabilizes the nanogap hotspots (Supplementary Fig. 6); however, chloride ions present (from all CB[$n$] crystals, $n = 5,6,7,8$) dissolve Au complexes leading to the destruction of SERS activity, as previously noted[52,53]. We thus alternate between Au oxidation in buffer, and reduction in 1 mM CB[5] + buffer (Supplementary Note 2, Supplementary Fig. 7). Stepping the potential to +1.5 V in buffer shows rapid simultaneous desorption of CB[5] and formation of Au oxide in the nanogap, occurring within 5 s. Switching to −0.80 V in CB[5] + buffer reduces the Au oxide and rapidly re-adsorbs CB[5] in the nanogaps within 2–3 s (Supplementary Fig. 8). Importantly, the SERS activity of the MLagg is stable over multiple electrooxidation/reduction repetitions, making this protocol promising for possible SERS substrate recycling (thus termed ReSERS: Re-cycling Scheme via Electro-oxidation and Re-Scaffolding).

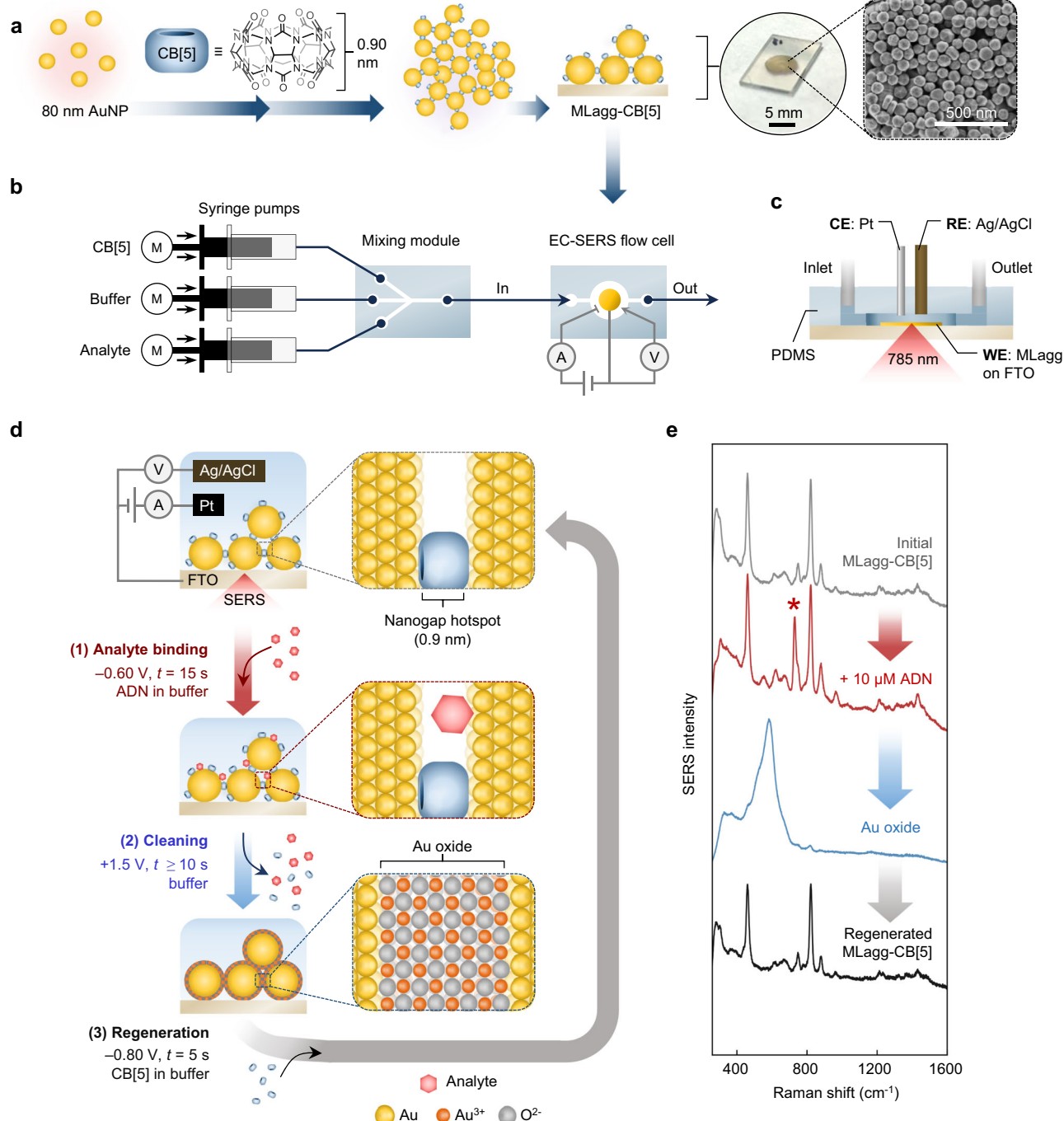

**Fig. 1 | Preparation and in-flow EC-SERS analyte detection, cleaning, and regeneration with MLagg-CB[5].** **a** Preparation of MLagg-CB[5] SERS aggregate from self-assembly of AuNPs with CB[5], followed by deposition onto solid support. Photo and scanning electron micrograph (SEM) show MLagg-CB[5] deposited on FTO-coated glass. **b** Schematic illustrates the integration of an MLagg-CB[5] into an EC-SERS flow system. **c** Cross-section of the EC-SERS flow cell (CE = counter electrode, RE = reference electrode, and WE = working electrode). **d** Schematic of in situ electrochemical SERS analyte detection and cleaning/regeneration protocol. Potentials are *vs* Ag/AgCl. **e** SERS spectra from: initial MLagg-CB[5] (grey), after detection of 10 μM adenine (ADN) (red), after oxidative cleaning step (blue), and after regeneration step (black). ADN peak at 732 cm$^{-1}$ is marked by asterisk. SERS spectra are collected with 1 s integration time and 1 mW 785 nm laser.

## Analyte detection and recycling repeatability

As an initial model analyte for recycling experiments, adenine (ADN) is selected since its potential-dependent binding and SERS spectra are well-studied[54,55], and tested in EC-SERS on roughened Ag electrodes[34]. At pH 7.0, ADN is predominantly neutral and binds to Au[56] however in electrolyte solutions, ADN competes with ions for binding sites in the nanogaps[57]. Negative potentials enhance ADN binding giving maximum SERS enhancement of the $\nu_{ADN} = 732$ cm$^{-1}$

peak with a step potential of −0.60 V (Supplementary Note 3, Supplementary Fig. 9). Once adenine binds to the MLagg-CB[5] hotspots, removal of this applied potential does not lead to adenine desorption, nor does application of moderately positive potentials (up to +0.60 V, before the onset of Au oxidation). Adenine is thus strongly bound and rinsing with buffer solutions (pH 2.0, 7.0, or 12.0), 0.1 M HCl, or 0.1 M NaOH does not lead to its desorption. Since adsorbed and is not removed with simple rinsing, it is an ideal

analyte to test in situ analyte detection, cleaning, and regeneration (Fig. 1d).

The SERS spectrum of the initial MLagg-CB[5] in the buffer-filled EC-SERS flow cell shows a clean MLagg with only the spectral fingerprint of the CB[5] molecular scaffold visible (Fig. 1e grey). To detect ADN, the analyte solution is injected into the flow cell using a syringe pump and a potential of −0.6 V applied for 15 s. A spectrum is then recorded at open-circuit potential (Fig. 1e red). After analyte binding, the nanogaps are cleaned by flowing buffer at a constant rate of 500 μL min⁻¹ while applying +1.5 V for >10 s. This oxidises/desorbs the ADN while forming Au oxide at the AuNP surface, seen directly in the SERS spectrum (Fig. 1e blue) as a broad peak ≈590 cm⁻¹ from the Au−O stretch[51,58]. Buffer flow for another 5 s after applying the oxidizing potential flushes out any decomposition products and desorbed analytes from the MLagg surface and sample chamber. After cleaning, the nanogap is regenerated by flowing 1 mM CB[5] in buffer and applying −0.8 V for 5 s, which rapidly reduces Au oxide while facilitating the binding of CB[5] onto the Au surface. The resulting SERS spectrum of the regenerated MLagg-CB[5] (Fig. 1e black) is near identical to the initial spectrum and shows no trace of the ADN $\nu_{ADN}$ peak, indicating how effectively the MLagg nanogaps are cleaned.

This in situ analyte detection and ReSERS cycle is then performed 30 times to evaluate its repeatability (Fig. 2a). To eliminate analyte signal variations due to spot-to-spot variations on the SERS substrate (Supplementary Fig. 10), the same substrate spot was probed for all cycles. Tracking the $\nu_{ADN}$ peak area every cleaning cycle shows that not only is the analyte successfully removed each time, but the MLagg is also identically regenerated, yielding a 5.5% relative standard deviation (RSD) between all repetitions (Fig. 2b). This variation in analyte detection is comparable to the 5.7% RSD previously observed when detecting and cleaning paracetamol from MLaggs using HCl rinsing over multiple cycles[9]. SERS mapping of the ADN signal at cycles 1, 10, 20, and 30 also shows that the analyte signal and regional uniformity of the substrate remain consistent, implying that all hotspots across the SERS substrate surface area are effectively and reproducibly regenerated (Supplementary Fig. 11). The SERS spectra from the regenerated MLagg are also consistent across the cycles (Fig. 2a, c), while SERS mapping shows the effective removal of the analyte across the probed area of the SERS substrate (Supplementary Fig. 11). In terms of the number of cycles demonstrated and % RSD, this level of recycling repeatability outperforms any reported method (Supplementary Table 1). We note that the MLagg can undergo at least 100 non-continuous analyte detection/ReSERS cycles, even if the MLagg is removed from the cell and dried intermittently. Under flow conditions, effective adhesion of the MLagg to the FTO-coated glass is required to ensure its robustness over continuous, prolonged use.

As a control, the same analyte detection, cleaning, and regeneration cycles are repeated on a new MLagg-CB[5] SERS substrate but without using a re-scaffolding molecule during the regeneration step (Fig. 2d,e). Specifically, after Au oxide formation, only buffer is pumped through the EC-SERS flow cell and −0.80 V is applied for 5 s to reduce the oxide. With this protocol, the analyte signal per cycle initially fluctuates and then gradually decreases (Fig. 2f), down to 12% of its strength by cycle 15. SERS maps of the analyte signal at cycles 1, 2, and 10 show similar variations in analyte signal across the probed area as well as a degradation in the local uniformity from 6% for cycle 1 to 12% and 29% RSD for cycles 2 and 10 respectively (Supplementary Fig. 12). A second cycling control using 1 mM KCl and buffer in the regeneration step characterizes the effect of chloride ions in the CB[5] solution, giving similar analyte signal fluctuations and a decrease to 21% by cycle 15 (Supplementary Fig. 13). A further control using MLaggs prepared from NaCl-aggregated AuNPs (without CB[5]) yields similar results (Supplementary Fig. 14).

## Morphology changes

The repeatable analyte detection achieved when CB[5] is added during the Au oxide reduction step suggests that the nanogaps are precisely regenerated. To investigate changes in morphology, the coupled plasmon modes of the MLagg are characterized using dark-field (DF) scattering spectroscopy (Fig. 3a), with the peak wavelength indicative of the gap size and refractive index[59]. In MLaggs the AuNPs are coupled via precisely-spaced nanogaps, resulting in strong plasmonic interactions between the nanoparticles[47,48]. Before cycling, the MLagg exhibits a chain mode at 745 ± 12 nm (shaded), set by the CB[5]-controlled gap size (0.90 nm). This coupled plasmon mode is highly sensitive to changes in gap size: assuming constant refractive index, a spacing increase by ≈0.1 nm results in a ≈20 nm blue shift[9]. After 10 and 30 cycles, we observe peak wavelength shifts to 741 ± 12 nm and 751 ± 8 nm respectively, indicating only minor effects on the plasmonic coupling. Tracking the plasmon mode after every regeneration step for the first 10 cycles confirms minimal shifts in peak wavelength relative to the initial MLagg (Supplementary Fig. 15). Representative SEM images of the MLagg-CB[5] before and after 10 and 30 cycles further evidence that the nanogaps are preserved (Fig. 3b).

In contrast, when CB[5] is omitted during the regeneration step, changes in nanogap morphology are observed as the MLagg is repeatedly electrooxidized and reduced. After the first cycle, the regenerated MLagg exhibits a ≈10 nm blue shift in peak wavelength and a decrease in scattering amplitude (Fig. 3c). While SEM imaging shows no obvious morphology changes (Fig. 3d, top), shifts in the coupled plasmon mode suggest partial sintering or nanoscale roughening at the nanogaps. The CB[5] SERS peaks after this first cycle decrease 30% and broaden (Fig. 2e, g), either from reduced CB[5] binding or reduced optical fields[60], while the SERS background more than doubles (Fig. 2g). Further rounds of Au oxide formation and reduction remove more CB[5] sites and induce more pronounced morphology changes. SEM images show formation of bridges between some neighbouring AuNPs after cycle 3 (Fig. 3d, centre, yellow arrows) with sintering of every nanogap after cycle 15 (Fig. 3d, bottom). The plasmon mode gradually blue shifts to ≈700 nm and broadens as the nanogaps collapse and sinter (Fig. 3c), coinciding with the disappearance of SERS (Fig. 2d–g). Similar morphology changes, plasmon mode shifts, and SERS decreases are observed with other controls (Supplementary Figs. 13, 14).

These data show that CB[5] is crucial for regenerating the nanogaps after oxidative cleaning. Time-resolved EC-SERS (Supplementary Figs. 6–8) shows that CB[5] rapidly binds to the Au facets to restabilise the nanogap hotspots as Au oxide is reduced. We propose that the rigid 0.90 nm high CB[5] molecule favourably binds to the two proximate AuNP facets[46], enabling the reproducible reconstruction of the sub-1 nm nanogap after Au oxidation. The CB[5] molecule acts as an effective molecular scaffold between the facets of neighbouring AuNPs, and its reintroduction prevents the nanogaps from collapsing and sintering. Nanoscale surface roughening visible in SEMs (Fig. 3b, d, Supplementary Figs. 13, 14) outside the nanogaps when cycling both with and without CB[5] is thus evidently very different from reconstructing the facet morphology inside the nanogaps with CB[5]. We note that EC-recycling is also possible with CB[6], but that CB[7] is harder to remove electrochemically from the nanogaps, while other scaffolding molecules are also inferior to CB[n].

## Analyte switching

To exemplify how this recycling removes all analytes from MLagg-CB[5] nanogaps, we alternate between detecting different molecules, thus robustly testing for SERS memory effects. If the first analyte is not effectively removed, its signal remains in the SERS spectrum as an interferent when the next analyte is detected. Incomplete analyte removal also results in competition for available binding sites in the SERS hotspots, which in turn affects detection sensitivity.

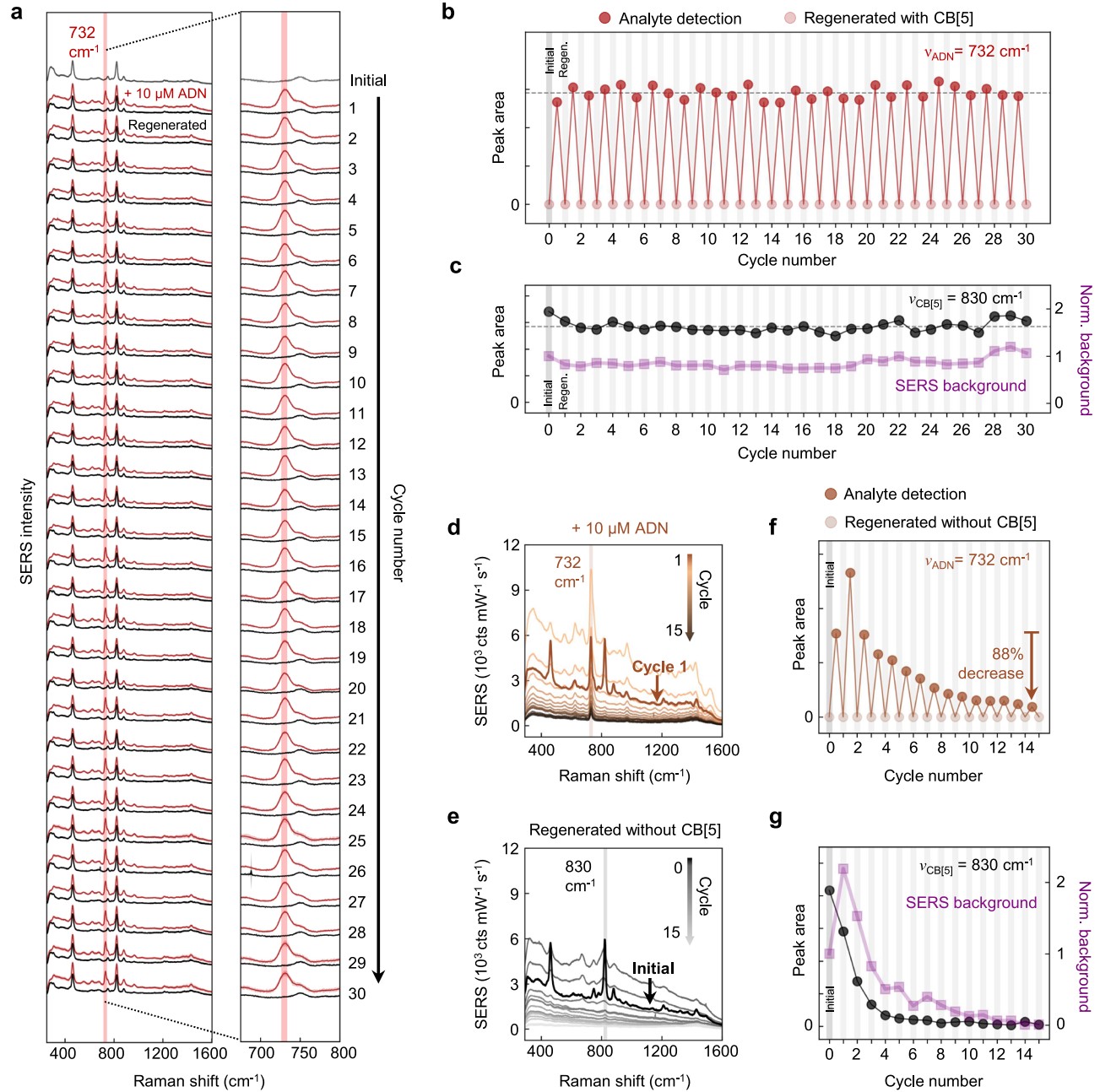

**Fig. 2 | Analyte detection, cleaning, and regeneration cycles with and without CB[5]. a** SERS spectra from 30 cycles of 10 μM ADN detection (red) and regeneration with CB[5] (black). Spectra are offset for clarity. **b** ADN peak areas ($\nu_{ADN}$ = 732 cm$^{-1}$) from the SERS spectra of each ADN detection and cleaning/CB[5]-regeneration cycle. Dotted horizontal line represents the average ADN peak area of all analyte detection cycles. **c** CB[5] peak areas ($\nu_{CB[5]}$ = 830 cm$^{-1}$, black circles) and integrated SERS background (purple squares) for the CB[5]-regenerated MLagg.

Dotted horizontal line represents the average CB[5] peak area of all regeneration cycles. **d** Overlaid SERS spectra from 15 cycles of 10 μM ADN detection and **e** after regeneration without CB[5]. A constant background was subtracted from all spectra to facilitate comparison across 15 cycles. **f** ADN peak areas ($\nu_{ADN}$ = 732 cm$^{-1}$) from the SERS spectra of each ADN detection and cleaning/regeneration cycle without CB[5]. **g** CB[5] peak areas ($\nu_{CB[5]}$ = 830 cm$^{-1}$, black circles) and integrated SERS background (purple squares) for the MLagg regenerated without CB[5].

Initially, we alternate between the detection of 100 μM ADN and 100 μM cytosine (CYT), selected since their Raman peaks do not overlap. They have similar electrochemical SERS enhancement at −0.60 V[34], allowing enhanced binding of both analytes simultaneously. Characteristic ADN and CYT ($\nu_{CYT}$ = 792 cm$^{-1}$) peaks are tracked for every detection/regeneration cycle (Fig. 4a, b), showing the complete removal of analytes each time and no detectable trace (within the ADN limit of detection ≈10$^{-7}$ M) of the initial analyte after analyte switching. Analyte quantification is repeatable to 4.6% RSD for ADN and 3.5% RSD

for CYT. A 50:50% mixture of 100 μM ADN and 100 μM CYT is also fully removed.

In the next set of analyte-switching experiments, ADN is alternated with thiols. Due to their strong interaction with Au, thiols readily displace most other adsorbates from Au surfaces, and are in turn very difficult to remove once bound[27]. Only a very limited number of SERS substrate recycling methods remove thiols, typically relying on harsh treatments such as the complete etching and redeposition of the metal layer[13,39], plasma or ozone treatment[18,27,61], or competitive binding with

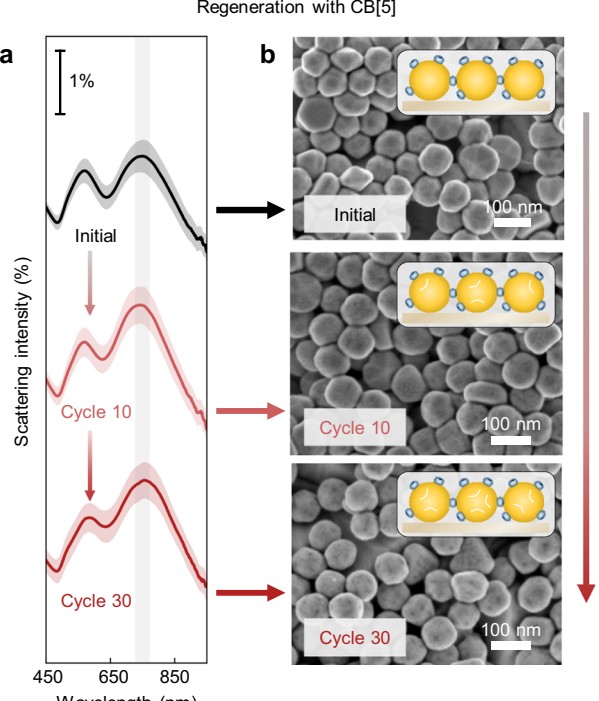

**Fig. 3 | Morphology changes with or without CB[5] during regeneration. a** DF scattering spectra and (**b**) representative SEM images of the MLagg-CB[5] before and after undergoing 10 and 30 cycles of analyte detection, cleaning, and regeneration with CB[5]. **c** DF scattering spectra and (**d**) representative SEM images of the MLagg-CB[5] throughout different cycles of analyte detection, cleaning, and regeneration without CB[5]. Yellow arrows in (**d**) point to bridges formed between neighbouring AuNPs. For the DF spectra, solid lines and shaded area represent mean and ±1 s.d. of *n* = 150 spectra obtained across the area of a MLagg-CB[5]. Grey line highlights initial chain mode peak wavelength. Spectra are offset for clarity.

NaBH$_4$[29,30]. Electrochemical oxidation-reduction cycling can remove thiol ligands from graphite-supported AuNP catalysts, but requires 25 cycles for complete thiol removal[42,43]. Thiols are sensitive to oxidizing conditions and at positive potentials near the oxidation of Au, surface-bound thiols (Au-SR) are oxidized to form R-SO$_2$[-62,63], which can be more readily removed from the surface with rinsing.

To evaluate the effectiveness of thiol removal using ReSERS, 100 μM of different aromatic thiols including 4-biphenylthiol (BPT), 4,4-biphenyldithiol (BPDT), 1,4-benzenedithiol (BDT), 2-naphthalenethiol (2-NT), and 4-mercaptopyridine (4-MPY) were each incubated with the MLagg-CB[5] for 1h and then cleaned. After analyte detection and cleaning by holding the potential at 1.5 V for 30 s under continuous buffer flow, each regenerated MLagg-CB[5] SERS spectrum shows no trace of the previously bound thiol (Fig. 4c), demonstrating its effective removal from the nanogaps. Before and after each thiol detection/ cleaning cycle, 100 μM of ADN was also detected to benchmark the cleaned/regenerated MLagg-CB[5]. This ADN calibration between each thiol detection was reproducible to 5.4% RSD (Fig. 4d), indicating highly effective cleaning and regeneration after thiol detection.

To simulate the continuous flow-injection analysis of various bioanalytes, different biologically significant compounds, including ADN, CYT, hypoxanthine (HYP), creatinine (CR), nicotinamide (NAM), paracetamol (PAR), norepinephrine (NEPI), tryptophan (TRP), nicotinic acid (NIA), and methylene blue (MB) at 100 μM each were sequentially detected after 15 s incubation and then cleaned from the same MLagg-CB[5] SERS substrate, with 15 s of ReSERS process between each analyte. SERS spectra from sequential analyte detection and cleaning/regeneration steps (Fig. 4e) prove the MLagg is effectively cleaned and regenerated. Calibration of ADN detection at the start and end of the series are within 6.0% in peak area, confirming the reliable regeneration of the SERS substrate throughout analysis of this test set. This demonstrates the capability to perform continuous

measurements of various analytes in aqueous flow systems, such as for flow-injection analysis or the sequential detection of fractionated chromatographic eluates.

## Quantitative analysis of ADN in biofluids

A key application of SERS substrate regeneration is analytical calibration and quantitative analysis. Most works reporting quantitative SERS analysis rely on using separate SERS substrates for every calibration standard and test sample[3–5]. Reliable quantitative analysis using this approach therefore hinges on the substrate-to-substrate reproducibility of the SERS substrate[8], and/or the use of the appropriate internal standards to normalize for other uncontrollable factors[64,65]. However, if a SERS substrate can be reliably cleaned and regenerated over multiple uses, then the same SERS substrate can be used for both the calibration and test samples. This greatly enhances the accuracy, reliability, and efficiency of the analysis by reducing the time and cost needed for SERS substrate fabrication, maximizing materials efficiency, and minimizing the uncertainty introduced by substrate-to-substrate variations.

For feasibility of absolute quantitative calibration using the same SERS substrate, ADN in buffer is again used as a model analyte. ADN calibration standards (0.1 μM to 100 μM) are sequentially detected and cleaned on the same MLagg-CB[5], extracting the SERS peak areas (Fig. 5a, b). ADN peaks are reproducible with average 5.3% RSD. Binding site competition is clearly observed as the CB[5] signal drops for higher ADN concentrations ≥5 μM (Fig. 5c). With such a repeatable system, analyte competition becomes feasible to quantify.

To test the viability of quantitative analysis on a real sample matrix using a single MLagg-CB[5] SERS substrate, the direct quantification of ADN in human urine is explored. Urine is a complex biofluid containing >4600 different metabolites at concentration levels spanning eleven orders of magnitude, from pM mM$^{-1}$ creatinine to mM mM$^{-1}$ creatinine

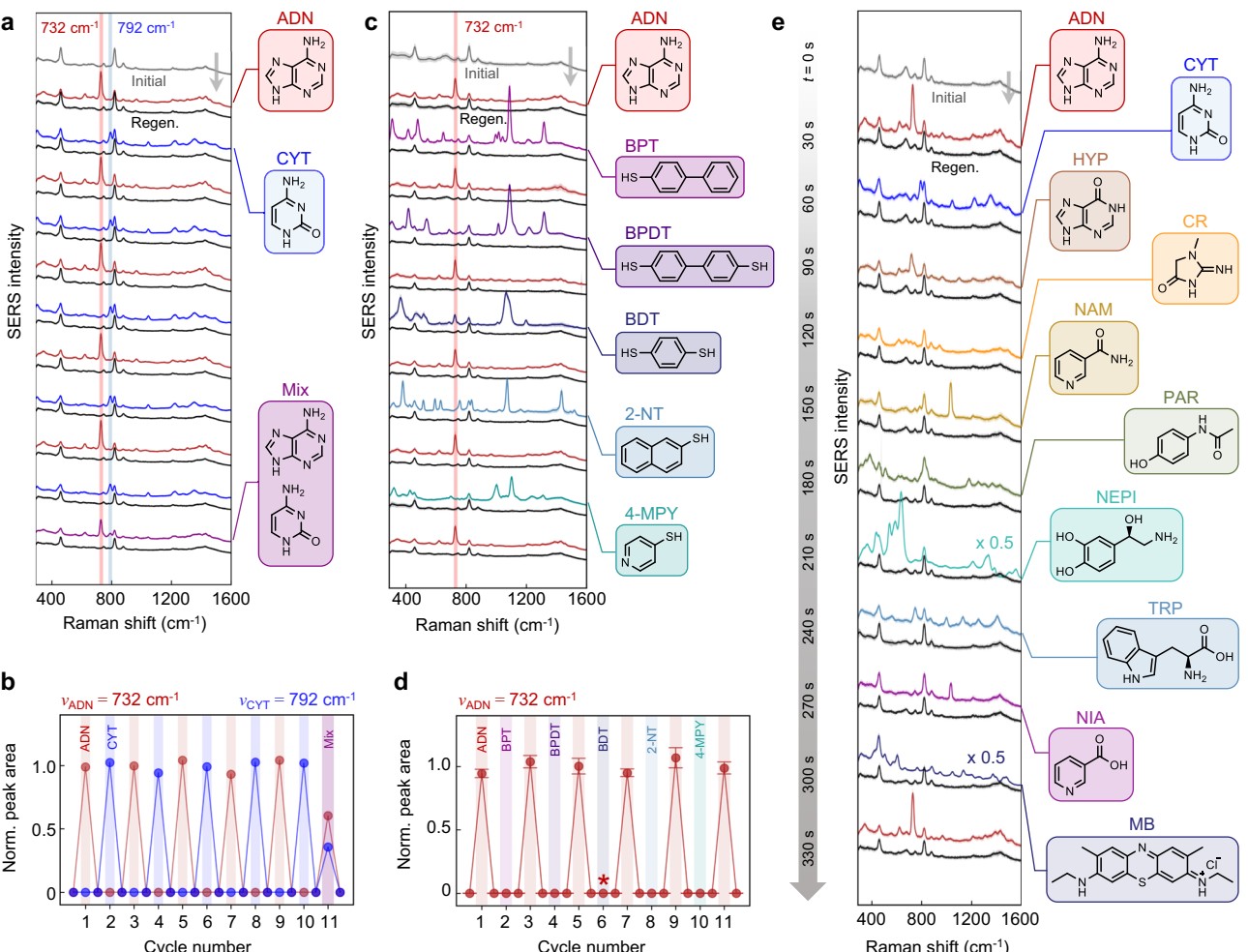

**Fig. 4 | In situ analyte switching. a** Sequential SERS spectra from alternating detection of 100 μM ADN (red) and 100 μM CYT (blue) with the corresponding regenerated MLagg-CB[5] (black). A 50:50% mixture of 100 μM ADN and 100 μM CYT (purple) is also tested. **b** Normalized peak areas of ADN ($\nu_{ADN}$ = 732 cm⁻¹, red) and CYT ($\nu_{CYT}$ = 792 cm⁻¹, blue) per detection-regeneration cycle. **c** Sequential spatially-averaged SERS spectra from alternating detection of 100 μM ADN (red) and 100 μM 4-biphenylthiol (BPT), 4,4-biphenyldithiol (BPDT), 1,4-benzenedithiol (BDT), 2-naphthalenethiol (2-NT), and 4-mercaptopyridine (4-MPY) with the corresponding regenerated MLagg-CB[5] (black). **d** Normalized peak area of $\nu_{ADN}$ per ADN/thiol detection-regeneration cycle. Peak areas are plotted as the mean with error bars representing ±1 s.d. of $n$ = 10 spectra obtained from different points across the MLagg-CB[5] area. For cycle 6 (marked by asterisk), alternative ADN peak at 970 cm⁻¹ is used due to overlap with BDT peak at 732 cm⁻¹. **e** Sequential SERS spectra from detection of a series of biological compounds: ADN, CYT, hypoxanthine (HYP), creatinine (CR), nicotinamide (NAM), paracetamol (PAR), norepinephrine (NEPI), tryptophan (TRP), nicotinic acid (NIA), methylene blue (MB), and ADN (red) again with the corresponding regenerated MLagg-CB[5] (black). Time ($t$) axis marks progress of both analyte detection and ReSERS (30 s between each detection cycle). All SERS spectra collected a1 s integration time and with 1 mW 785 nm excitation laser.

(normalized to creatinine levels as conventional)[66]. Typical levels of urinary ADN in adults occur at mid-range concentrations of 1.4–5.1 μM mM⁻¹ creatinine, or 18–66 μM given average creatinine concentrations of 13 mM[66]. Other major molecular components of urine such as hippuric acid, citric acid, urea, creatinine, and uric acid occur at mM levels[66]. Direct analysis of a complex sample therefore exposes SERS substrates to a broad range of unknown components of varying concentrations that often result in substrate fouling. Our aim is to mitigate this using the ReSERS cleaning technique since it is proven capable of stripping even thiols.

The many unknown components in real sample matrices pose not only a challenge to SERS substrate cleaning, but also for the quantitative analysis itself. Due to the complexity of biological matrices, ADN is not usually determined directly, but typically requires chromatographic separation prior to detection[67]. As SERS relies on the interaction of the analyte with the nanogap for detection, direct analysis of a complex sample matrix such as urine inevitably leads to the simultaneous competitive binding of molecular components at the SERS

hotspots, thereby limiting the availability of binding sites, and reducing the analysis sensitivity. Calibration standards must thus be prepared in a background that matches that of the sample (e.g., ionic strength, pH, background components) to include interactions of the SERS substrate and analyte with the background matrix. However, for complex matrices such as urine, this can be difficult and/or impractical to prepare.

To address this challenge, we opt to use the standard addition method to account for matrix interactions in direct analysis[65]. For ADN determination in urine, a test sample of human urine is prepared (see Methods) containing 50.0 μM ADN (Supplementary Fig. 16) and with a series of standard addition calibrators of [ADN]$_{added}$ = 0, 12.5, 25.0, and 37.5 μM (Fig. 5d). Using the same substrate throughout, the SERS spectrum of each standard is measured after applying the ADN enhancement potential of −0.60 V, followed by cleaning and regeneration (Supplementary Fig. 17). The triplicate measurements at $\nu_{ADN}$ of each standard are repeatable, with 1.0-3.9% RSD for each (Fig. 5e). To ensure linearity, the standard [ADN]$_{added}$ = 37.5 μM is excluded from

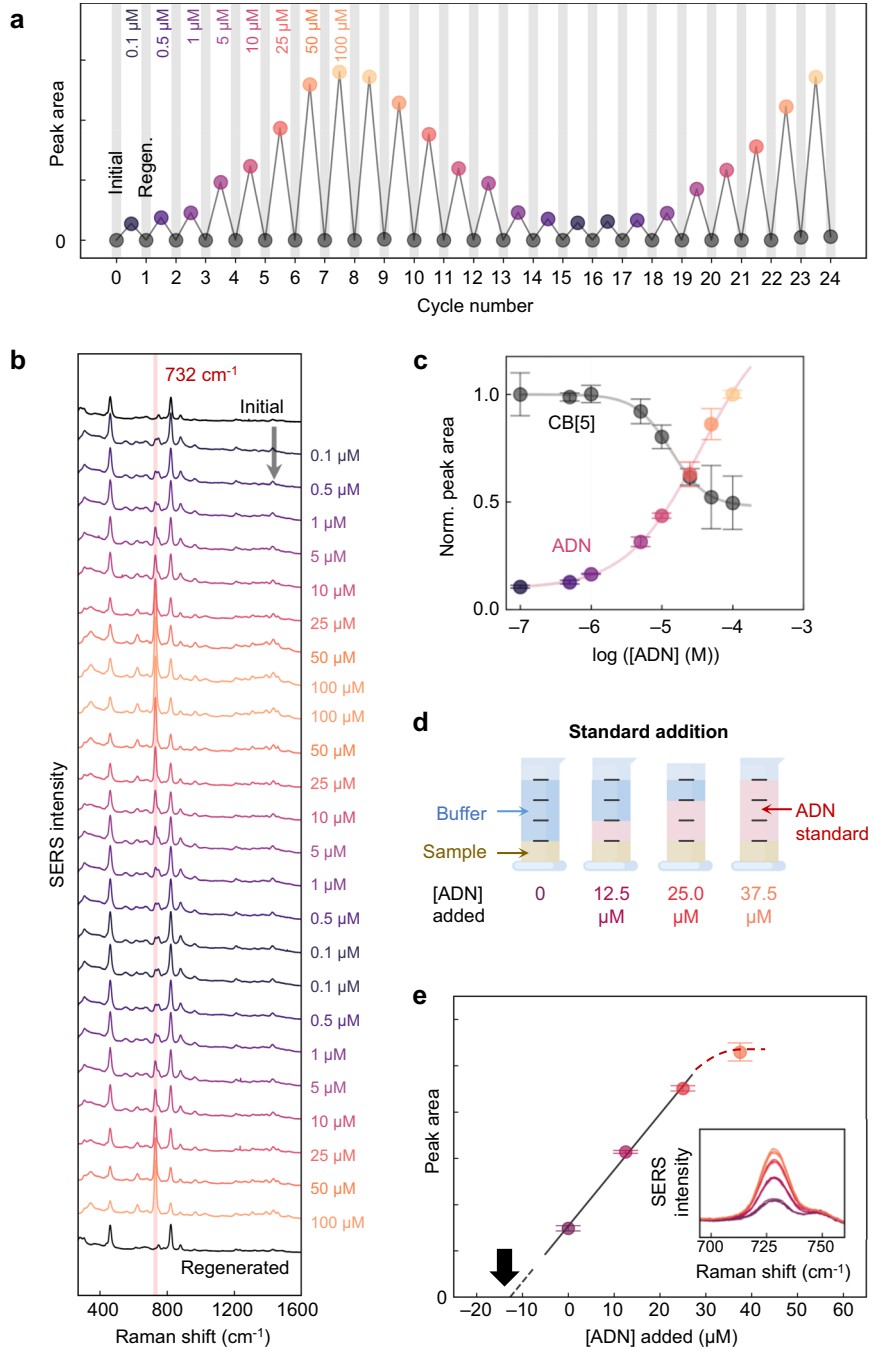

**Fig. 5 | Quantitative analysis of ADN. a** ADN peak areas at $\nu_{ADN}$ from sequentially measured calibration standards (0.1 to 100 μM) in 50 mM potassium phosphate buffer (pH 7.0) and regenerated MLagg-CB[5] (black circles). **b** Sequential SERS spectra from ADN calibration standards measured from the same MLagg-CB[5] SERS substrate. SERS spectra collected at 1 s integration time and 1 mW 785 nm laser. **c** Relative peak areas of CB[5] ($\nu_{CB[5]}$ = 830 cm$^{-1}$) and ADN ($\nu_{ADN.}$ = 732 cm$^{-1}$) vs [ADN] (log scale). Peak areas are plotted as the mean with error bars representing ±1 s.d. from $n$ = 3 measurements of an ADN calibration standard using the same MLagg-CB[5] SERS substrate after multiple ReSERS cycles. Solid lines are fits. **d** Schematic of the standard addition analysis of ADN in urine. **e** ADN peak area vs added [ADN] standard. Points are plotted as mean ±1 s.d. from $n$ = 3 measurements of an ADN calibration standard using the same MLagg-CB[5] SERS substrate after multiple ReSERS cycles. Grey solid line is linear fit, dashed grey line is linear extrapolation. Black arrow points to the extrapolated $x$-intercept. Red dashed line highlights the trend towards non-linearity at higher concentration. Inset shows close-up of ADN peak from SERS spectra of calibration standards.

the calibration fit since analyte saturation is evident ($R^2$ = 0.996, $p < 0.05$). Extrapolating (Fig. 5e, black arrow) and accounting for dilution returns the ADN concentration in the test sample as 50.8 μM (with error ±6.0 μM at 95% confidence), and 101.6% recovery.

The high accuracy of ADN determination in urine using the same MLagg-CB[5] indicates that our SERS substrate recycling method indeed mitigates substrate fouling, enabling reliable continuous reuse

of the SERS substrate for the entire standard addition assay. While standard addition was previously demonstrated for direct quantification of uric acid[68,69] and moxifloxacin[70] in urine, new SERS substrates were needed for every measurement, generating high levels of waste, while substrate-to-substrate variations compromise the accuracy and reproducibility (6–9% error, 8–25% RSD reported). By contrast, using single MLagg-CB[5] substrates coupled with effective recycling offers

multiple advantages, including cost savings, improved efficiency, enhanced precision, ease of comparison, and the capability for integration. These benefits contribute to more accurate and reliable analytical calibration, supporting high-quality and high-throughput quantitative analysis across various applications, including environmental monitoring, industrial quality control, and medical diagnostics.

## Discussion

SERS has been enormously hampered by the difficulty of preserving the metallic nanogaps that deliver the crucial signal enhancements during its practical use for analyte sensing. Here, we demonstrate an in situ electrochemical SERS substrate recycling method that addresses the challenge of substrate degradation and limited reusability. Compared to other previously reported recycling methods, we demonstrate a two-step electrochemical process that rapidly removes a broad range of analytes (all we have tested so far) and reproducibly regenerates the hotspots of the SERS substrate through the unusual and rapid reintroduction of a nanogap-stabilizing scaffold molecule. The ability to maintain consistent SERS enhancements after multiple cycles of reuse showcases the robustness of our method and its potential to provide reliable analytical results over extended periods. This can significantly impact the development of highly sensitive and reproducible SERS platforms for diverse applications such as trace-level molecular detection, biosensing, and environmental monitoring.

Several key conditions are required for this cleaning. Initial gaps between the AuNPs need to be on the few nm scale, so that oxidation reliably and completely plugs them with Au oxide. The MLagg (or equivalent plasmonic nano-geometry) must be fully electrically contacted to act as the working electrode and possess low enough tortuosity that the electrical double layer accesses each nanogap. For instance, we find that stacking six layers of the MLagg (which tunes the resonant modes into the mid-infrared) is also found to be successfully cleaned. Chloride ions must be eliminated from the oxidation step as they evidently interfere with the Au oxide formation. Details of the mechanism for the nanoscale process will need to confirm the crystal structure and stoichiometry with depth of the Au oxide that plugs the nanogaps, as clearly seen in the SERS and scattering spectra. The variety of SERS peaks ≈590 cm$^{-1}$ that make up the Au oxide signature have been matched to different positions of oxygen atoms in the upper and lower Au atomic layers[58]. An improved understanding should also show how the CB[$n$] molecules interact with the Au oxide as it progressively reduces and recreates the separated Au facets that elicit extremely strong SERS signals for analytes. Although we find that other molecules can perform this re-scaffolding (as in ref. 9), their cleaning repeatability is inferior to CB[5] so far and sintering is progressively observed. An account is also sought for the capability of analytes to diffuse into the SERS hotspots once the CB[$n$] has re-scaffolded the nanogap. The combination of electrochemical potentials at the nanoscale surfaces, variable oxidation states, labile hydrogen bonding between organic and metal oxide, and confinement at the nanoscale, all highlight the complexity of the ReSERS process. Because the chemistry of Au is rather unique, it is not clear how well this cleaning can be controlled on other SERS-active metals such as Ag and Cu. However, we suggest that analogous approaches may indeed also be successful. We also note this treatment will equally enhance the repeatable performance of sensors for surface-enhanced infrared absorption (SEIRA).

To ensure the successful integration of this method for real-world SERS applications, further studies will be directed towards optimization, validation, and mechanistic investigations. Specifically, optimizing the electrochemical parameters and electrolyte composition to achieve the highest recycling efficiency without compromising long-term substrate integrity is required. Optimization towards minimizing the time needed for the two-step ReSERS process is ongoing, to improve its suitability for rapid on-line detection, while additional rigorous studies involving more complex multi-analyte samples will be explored to validate its performance in real-world scenarios. Further work is underway to understand the nanogap regeneration mechanism, which can provide insights on achieving more rapid implementation and longer-term stability, as well as extending the applicability of the protocol to other SERS substrate metals and scaffolding molecules. Systematically addressing these research avenues can lead to the seamless integration of electrochemical recycling with SERS platforms in microfluidic compact sensor systems, contributing to the enhanced cost-effectiveness, practicality, and reliability of SERS-based analysis.

## Methods
### Materials

All chemicals were used as received. Citrate-stabilized 80 nm AuNPs (optical density 1.0 at 555 nm) were purchased from BBI Solutions. Analytical-grade chloroform (≥ 99.8 %) was obtained from Merck. HCl (37%) and methylene blue hydrate (MB) (≥96%) were from Fisher Scientific. NaCl (≥ 99%), K$_2$HPO$_4$ (≥98%), and KH$_2$PO$_4$ (≥98%) were from Alfa Aesar. Cucurbit[5]uril hydrate (≈ 20% water), adenine (ADN, ≥99%), cytosine (CYT, ≥99%), 1,4-benzenedithiol (BDT, ≥99%), 2-naphthalenethiol (2-NT, ≥99%), 4-biphenylthiol (BPT, ≥97%), 4,4-biphenyldithiol (BPDT, ≥95%), 4-mercaptopyridine (4-MPY, ≥95%), hypoxanthine (HYP, ≥99.0%), creatinine (CR, ≥98%), nicotinamide (NAM, ≥98%), paracetamol (PAR, ≥99%), (±)-norepinephrine hydrochloride (NEPI, ≥97%), L-tryptophan (TRP, ≥98%), nicotinic acid (NA, ≥98%), and 96% ethanol (EtOH) were obtained from Sigma-Aldrich. Lyophilized pooled human urine controls (Level II for biogenic amines, Ref 8821, Lot 1066) were from ClinChek Controls. Polydimethylsiloxane (PDMS) was prepared using a SYLGARD 184 kit from DOWSIL (Dow Silicones). Fluorine-doped tin oxide (FTO)-coated glass slides (TEC 10) were purchased from Ossila Ltd and were cleaned and cut to 10 × 15 mm$^2$ slides prior to use. All aqueous solutions were prepared using deionized (DI) water (>18.2-MΩ cm$^{-1}$) from a Purelab Ultra Scientific water purification system.

### Multilayer aggregate preparation

MLagg SERS substrates were prepared[9,47] by first placing 500 $\mu$L of 80 nm AuNP with an equal volume of chloroform in an Eppendorf tube. Aggregation of the AuNPs was initiated upon the addition of 50 $\mu$L of 1 mM CB[5] or 1 M NaCl and was facilitated by vigorous shaking for 1 min. The aggregates were then allowed to settle at the aqueous-organic interface. Excess ligands and salts were removed by replacing the aqueous supernatant with fresh DI water. This washing step was repeated three times. The aggregates were then transferred by carefully decreasing the volume of the aqueous phase to ≈5 $\mu$L and depositing the droplet onto a pre-cleaned FTO glass slide. Once deposited, the MLagg was air dried, rinsed with DI, and dried with compressed N$_2$. A schematic of the full protocol is summarized in Supplementary Fig. 1a.

To remove the remaining native AuNP ligands from the MLagg surface, all surface ligands were stripped with oxygen plasma cleaning using 90% RF power and 30 cm$^3_{STP}$ min$^{-1}$ (Henniker Plasma, HPT-100) for 45 min. The desired scaffolding ligand (e.g., CB[5]) was then introduced by incubating the plasma cleaned MLagg in a 1 mM ligand solution prepared in 0.5 M HCl (Supplementary Fig. 1b−e). After 5 min, the MLagg was rinsed with DI and dried with N$_2$. Alternatively, initial MLagg cleaning was also performed with in situ electrochemical cleaning using the regeneration protocol described below (Supplementary Fig. 18).

### SERS and dark-field measurements

SERS measurements were recorded on a custom-built Raman set-up (Supplementary Fig. 4) using a 785 nm diode laser (Matchbox) set at ≤1 mW power. Excitation and collection were performed through an

Olympus LUMPlanFl/IR ×40 W NA 0.80 water-immersion objective (in inverted configuration), and spectra were recorded by an Andor Newton 970 EMCCD camera coupled to a Shamrock 168 spectrometer with 1 s integration times.

SERS mapping measurements were taken on a commercial Raman instrument (Renishaw inVia) with 1 s integration times, 785 nm excitation (laser line profile) and 2.1 mW laser power using a ×20 NA 0.40 objective. Map scans were taken over a 465x330 $\mu m^2$ region over a $31 \times 11$ grid with $15 \times 30$ $\mu m^2$ spacings.

DF scattering spectra were recorded on a modified Olympus BX51 with an Ocean Optics QE-Pro spectrometer with 0.5 s integration time. Excitation and collection were performed through an Olympus MPLanFL N ×20 BD NA 0.45 objective. DF scattering spectra of MLagg samples were collected over a $600 \times 400$ $\mu m$ area (in a $10 \times 15$ point grid) and then averaged. A white light scattering target (Labsphere) was used as a reference to normalize white light scattering.

## SEM measurements

SEM measurements of MLaggs deposited on FTO-coated glass slides were taken on a FEI Philips Dualbeam Quanta 3D SEM (dwell 3–10 $\mu s$, HV 2 kV, current 50 pA, and ≈2.0 mm WD) or on a FEI Helios NanoLab 650 SEM (dwell 100 $ns^{-1}$ $\mu s$, HV 20 kV, current 100 pA and ≈4.0 mm WD). Magnification ranged from 80,000–200,000x.

## SERS enhancement factor calculation

The SERS enhancement factor of MLagg-CB[5] was calculated based on the SERS and Raman signal of CB[5] (Supplementary Fig. 2) normalized to an estimated number of probed molecules. The CB[5] SERS spectrum was measured on an MLagg-CB[5] using 1 s integration time and 2.1 mW 785 nm laser excitation. The Raman spectrum of a 2 mM CB[5] solution in water was taken with 200 s integration time and 127 mW 785 nm laser excitation. All measurements were taken using a Renishaw inVia with a ×20 NA 0.40 objective.

To estimate the number of CB[5] molecules probed per SERS measurement, a diffraction-limited laser spot size of 1 $\mu m^2$ was assumed. The number of nanogaps per $\mu m^2$ and the average nanogap diameter were estimated from SEM imaging, while the surface coverage of CB[5] molecules was based on previously reported X-ray photoelectron spectroscopy measurements[47]. From here, the number of CB[5] molecules bound in nanogaps per 1 $\mu m^2$ probed ($N_{SERS}$) was calculated

$$N_{SERS} = \rho_{CB[5]} \cdot (D_{nanogap}/2)^2 \cdot n_{gaps} = 40,050 \text{ CB[5] molecules} \quad (1)$$

where $\rho_{CB[5]}$ = surface density of CB[5] molecules on MLagg-CB[5] = $9 \times 10^{17}$ $m^{-2}$, $D_{nanogap}$ = average lateral width of nanogap = 30 nm, and $n_{gaps}$ = number of gaps within laser focus = 200 on average. The number of CB[5] molecules probed via Raman ($N_{Raman}$) in a solution of 2 mM CB[5] was then estimated,

$$N_{Raman} = c \cdot N_A \cdot D^2 z = 6,022,000 \text{ CB[5] molecules} \quad (2)$$

where $c$ = 2 mM concentration of solution for Raman, $N_A$ = Avogadro's number, $D$ = 1 $\mu m$ diameter of Raman excitation laser spot, $z$ = 5 $\mu m$ depth of focus. Comparing the normalized Raman peak intensity in counts $mW^{-1}$ $s^{-1}$ with a 785 nm excitation laser for the CB[5] band at 830 $cm^{-1}$ (Eq. 3) yielded $1.0 \times 10^6$, which is in line with typical SERS enhancements for a variety of substrates and non-resonant SERS molecules.

$$\text{SERS EF} = \frac{I_{SERS}/N_{SERS}}{I_{Raman}/N_{Raman}} = \frac{\left(\frac{1,400 \text{ counts mW}^{-1}\text{s}^{-1}}{40,050 \text{ molecules}}\right)}{\left(\frac{0.2 \text{ counts mW}^{-1}\text{s}^{-1}}{6,022,000 \text{ molecules}}\right)} = 1.0. \times 10^6 \quad (3)$$

## EC-SERS flow set-up

A miniaturized EC-SERS flow cell was designed and fabricated to accommodate a standard three-electrode electrochemical system: a leakless Ag/AgCl reference electrode (LF-1-45 from Innovative Instruments Ltd), a Pt wire (Sigma-Aldrich) counter electrode, and a removable MLagg SERS substrate on FTO-coated glass as the working electrode (Fig. 1c). The internal volume of the flow cell was 26 $\mu$L. The EC-SERS flow cell and a three-inlet/one-outlet mixer module were fabricated with PDMS using 3D-printed moulds. The EC-SERS flow cell was sealed and mounted onto the stage of an inverted Raman set-up using custom 3D-printed holders and bases.

Custom-built syringe pumps were used to control the flow of solutions (buffer, CB[5] in buffer, and analyte in buffer) through the EC-SERS flow cell (Supplementary Fig. 4). Electrochemical measurements were conducted using a portable potentiostat (CompactStat) from Ivium Technologies. All potentials were referenced to the Ag/AgCl reference electrode. The syringe pumps, electrochemical measurements, and SERS spectra collection were all controlled and synchronized with Python scripts.

## Analyte detection

Aqueous analyte solutions were prepared in a background electrolyte of 50 mM potassium phosphate buffer (pH 7.0, 0.5 mS $cm^{-1}$ conductivity) and injected into the EC-SERS flow cell. Under static conditions, an electrochemical enhancement potential (−0.60 V) was applied where applicable for 15 s. SERS spectra were then collected at open circuit potential.

For the detection of thiols, the MLagg was removed from the EC-SERS flow cell and immersed in a 100 $\mu$M thiol solution in EtOH for 1 h. Afterward, the MLagg was rinsed with EtOH and dried with a stream of $N_2$. The MLagg was then reinstalled into the EC-SERS flow cell for cleaning, regeneration, and reference calibrations of ADN detection. Since the MLagg was periodically removed from the cell, it was difficult to probe precisely the same spot on the MLagg surface, so spectra were taken over multiple random spots across the substrate ($n = 10$) and averaged. Variations in the spectra in this case thus also include spatial variation across the MLagg.

## Cleaning and regeneration

To clean and regenerate the MLagg, 50 mM potassium phosphate buffer (pH 7.0) was pumped into the EC-SERS flow cell and a potential of +1.5 V vs Ag/AgCl was held for 5–60 s under continuous buffer flow (flow rate = 500 $\mu$L $min^{-1}$). For initial MLagg cleaning, 60 s was typically required, while for analyte cleaning, 15–30 s was sufficient. After cleaning, 1 mM CB[5] in 50 mM potassium phosphate buffer (pH 7.0) was pumped into the flow cell. Under static conditions, a potential of −0.80 V vs Ag/AgCl was held for 5 s. Buffer was then flushed into the cell to remove excess scaffold molecules. If traces of previously detected analyte were evident from the SERS spectrum, another round of cleaning/regeneration was conducted.

During detection/cleaning cycling experiments, solution syringes were refilled every 12–15 cycles. The flow system was allowed to stabilize before measurements were resumed. Care was also taken to record the SERS spectra from the same substrate spot throughout the cycling experiments. However, the continuous operation of the flow system was occasionally halted to replenish the water droplet on the water immersion objective, which tended to dry over prolonged use. Since this required moving the EC-SERS flow cell, some variation in the probed substrate spot can occur.

## Standard addition

For direct ADN detection in urine, a test sample was prepared from a pooled human urine control matrix, which initially contained no

detectable trace of ADN (Supplementary Fig. 16). The test sample was prepared by dissolving the lyophilized matrix in 50 mM potassium phosphate buffer and adding ADN standards for a final concentration of 50 μM ADN. Calibration standards were prepared by taking equal volume aliquots of the test sample and adding different volumes of 100 μM ADN standard and 50 mM potassium phosphate buffer. The final concentrations of added ADN for each standard were 0, 12.5, 25.0, and 37.5 μM. The final dilution factor of the test sample in all the standards was 4.

### Data analysis

SERS spectra are presented here with minimal data processing, except for background correction to eliminate the broad glass background signal centred at 1400 cm$^{-1}$ that arises from back-side optical measurements. Analyte peak areas were determined by iteratively fitting a polynomial to correct for the SERS background, followed by fitting Gaussian curves to the narrow analyte peaks of interest. To determine the peak wavelength of the MLagg coupled plasmon mode from DF scattering spectra, Gaussian curves were fitted for each DF spectrum. The peak wavelength was determined from the centre of the fitted Gaussian.

### Reporting summary

Further information on research design is available in the Nature Portfolio Reporting Summary linked to this article.

## Data availability

The data that support the findings of this study are available from the Cambridge Open Data archive[71] and from the corresponding author upon request.

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

## Acknowledgements

The authors greatly appreciate helpful comments from many colleagues including Oren Scherman, Luis Liz-Marzán, and Duncan Graham. The authors acknowledge financial support from the European Research Council (ERC) under Horizon 2020 research and innovation programme PICOFORCE (Grant Agreement No. 883703), and POSEIDON (Grant Agreement No. 861950) and from the EPSRC (Cambridge NanoDTC EP/L015978/1, EP/L027151/1, EP/X037770/1). S.M.S.-T. is supported by the University of Cambridge Harding Distinguished Postgraduate Scholars Programme. S.M.S.-T., D.-B.G., and N.S. acknowledge support from EPSRC Grant EP/L015889/1 for the EPSRC Centre for Doctoral Training in Sensor Technologies and Applications, and from AstraZeneca (MedImmune Ltd). M.N. is supported by a Gates Cambridge fellowship (OPP1144). B.d.N. acknowledges support from the Royal Society (URF/R1/211162).

## Author contributions

S.M.S.-T. and J.J.B. conceived and designed the experiments. EC-SERS flow set-up was developed by S.M.S-T., D.-B.G., and G.K., with input from M.N., E.W., and B.d.N. S.M.S-T. performed fabrication and spectroscopic experiments with input from D.-B.G., G.K., M.N., E.W., and B.d.N. S.M.S.-T. and E.W. analysed the data with input from M.N. Sample fabrication was aided by M.N. and E.W. SEM image collection was carried out by N.S. and A.R. S.M.S.-T. and J.J.B. wrote the manuscript with input from all authors.

## Competing interests

The authors J.J.B., S.M.S.-T., D.-B.G., M.N., and E.W. declare the following competing interests: filed patent, Surface-enhanced spectroscopy substrates, UK 2304765.7, 30/3/2023. The authors G.K., N.S., A.R., and B.d.N. declare no competing interests.
