## [Peer Review File · Nature Communications]

In-situ electrochemical regeneration of nanogap hotspots for continuously reusable ultrathin SERS sensorsReviewers' Comments:

Reviewer #1:

Remarks to the Author:

This is a very interesting article on advancing SERS, a powerful analytical technique that allows chemical compounds to be identified and quantified via fingerprints and provides structural information. Currently, commercial use is limited due to significant drawbacks such as poor reusability of SERS targets, poor reproducibility, limited connectivity with online methods and low universality in terms of compounds that can be measured.

In their very well written paper, Sibug-Torres et al. present an electrochemical method to overcome the memory effect of Au-SERS targets in a reproducible and easy-to-use manner. A detailed description of the various system parameters and influencing factors underpins their research. One of the strengths of this work is the detailed and comprehensive and well-understood description of their system and the demonstration of its applicability to non-model SERS compounds. They also demonstrate the applicability of their approach to real-world applications such as the analysis and quantification of adenosine in urine.

Due to the high importance and originality of their research towards making SERS available for commercial applications and the extensive and detailed description, it is recommended that this paper be accepted after incorporating the following edits:

Main article:

Abstract: The abstract should include some more features of the presented system: Which scaffold molecule is used. How long did it take to regenerate the target? How reproducible was the amplification factor (standard deviation) and what potential was used? This information should be obvious to any reader at first glance. The gain factor of the SERS substrate used should also be added.

Introduction: The current limitations of SERS in terms of target reusability are listed. However, one of the most important limitations is not mentioned: the poor combinability with on-line methods, while later (page 7, last line) it is said that the approach would be suitable for the detection of chromatographic eluates.

In this context, a high regeneration speed would be essential, but this information (how fast it goes) is not sufficiently clearly presented. I would suggest to state more explicitly how fast the two-step regeneration process is. Here it would be useful to also include this information in Figure 4 in the form of an x-time axis.

a few minor comments:

Page 1, line 46: The claim that all the methods mentioned lead to morphological changes on the surface of the substrate is not substantiated.

Page 2 line 8: The word "scaffolding" is still unclear in the text and should be defined earlier to simplify the understanding of the text.

Page 2 line 24: The benefits mentioned should be defined in the text.

The experimental setup shown in Figure S3 should be included in the main article.

Page 10: The standard addition procedure is explained in great detail. All related paragraphs can be deleted as the standard addition procedure is common when dealing with samples containing complex matrices.

Please include a picture of the optical setup used in the SI.

Page 12: Analytical detection: Please indicate the conductivity of the buffer used.

Supplementary information:

S5 Graph b should be shown as a spectroelectrochemistry graph. The Raman intensity of each wavelength should be shown in relation to the applied potential. An example would be Figure 2 in the following publication: [10.1016/j.jelechem.2020.114743](https://doi.org/10.1016/j.jelechem.2020.114743)

Page 6: At the end of the first paragraph it is mentioned that CB5 is desorbed in favour of anions in the EDL when a positive potential is applied. Is it possible that CB5 undergoes an electrochemical reaction and is reduced before desorption?

Page 8: Please add at which wavenumbers the phosphate ion peaks are visible.

Page 11: I recommend plotting this diagram as a spectrovoltammogram, as previously described

Reviewer #2:

Remarks to the Author:

Reviewer #3:

Remarks to the Author:

In this work, an in situ electrochemical SERS hot regeneration scheme (ReSERS) was proposed for the continuous reuse of thin film gold nanoparticles (AuNP) SERS substrates. It is challenging to realize the reuse of SERS substrate that is very important in the SERS field. This work shows that the SERS substrate demonstrates excellent repeatability and regeneration cycles 30 times, allowing direct quantitative analysis of complex samples without substrate contamination. It provides continuous high-throughput SERS measurements and has wide applications in health, environment, industrial processes, and quality monitoring. I suggest accepting it after dealing with the following issues.

1. The electrochemical method has been used to regenerate the SERS substrate. Please add more discussions about the differences between the previous work and highlight the advantages of this work. I suggest giving more discussion about the mechanism of the action of cucurbit [5] uril (CB [5]) ligands in the work.
2. In this work, the authors state that the gold can be oxidized during the electrochemical redox reactions. Will the morphology change after the 30 times cycles? It would be good to characterize the nanostructures before and after the cycles.
3. It is well known that the thiols have a high affinity with gold. When exposed to thiols, is it possible that the thiol molecules may replace the CB in the nanovoids?
4. Considering the solid SERS substrate, what about the uniformity of the SERS measurements? Does the SERS substrate can keep uniformity after the cycling tests? To give the SERS mapping data would be more convincing.
5. The article only mentions renewable data with 30 repetitions. Is it the maximum limit number of repetitions for this base?
6. The relevant parameters of SERS measurements should be supplemented.
7. The related work maybe help to strengthen the research background, such as, Recyclable Surface-Enhanced Raman Scattering Substrate-Based Sensors, ACS Sustainable Chemistry & Engineering, (2023), DOI: 10.1021/acssuschemeng.2c05291, and Moreover, please modify the reference format carefully, i.e. the reference format needs to be unified according to the journal format requirements.

Response to reviewers:

We are delighted that the reviewers find our '*interesting*', '*very well written paper*', is of '*high importance and originality*' (Reviewer #1), recognise that '*one of the strengths of this work is the detailed and comprehensive and well-understood description of their system and the demonstration of its applicability to non-model SERS compounds*' (Reviewer #1), and agree that '*the SERS substrate demonstrates excellent repeatability and regeneration cycles*', which is '*very important*' and '*has wide applications*' (Reviewers #2&3). The reviewers ask for a few further clarifications and make valuable recommendations to improve the manuscript, which are all addressed below, in addition to providing extensive additional supporting data.

Reviewer #1:

1. The abstract should include some more features of the presented system: Which scaffold molecule is used. How long did it take to regenerate the target? How reproducible was the amplification factor (standard deviation) and what potential was used? This information should be obvious to any reader at first glance. The gain factor of the SERS substrate used should also be added.

> This is a helpful set of additions. We now include the key features and performance of our system in the abstract. Specifically, we note the time required for regeneration (≥ 15 s), the scaffold molecule used (cucurbit[5]uril), the potentials used for cleaning (+1.5V vs Ag/AgCl) and regeneration (-0.80V), and the reproducibility in terms of the relative standard deviation between repeated analyte detection cycles. We also include a calculation of the enhancement factor of the MLagg-CB[5] SERS substrate (conservatively estimated as $\sim 10^6$), detailed in the Supplementary Information.

2. Introduction: The current limitations of SERS in terms of target reusability are listed. However, one of the most important limitations is not mentioned: the poor combinability with on-line methods, while later (page 7, last line) it is said that the approach would be suitable for the detection of chromatographic eluates. In this context, a high regeneration speed would be essential, but this information (how fast it goes) is not sufficiently clearly presented. I would suggest to state more explicitly how fast the two-step regeneration process is. Here it would be useful to also include this information in Figure 4 in the form of an x-time axis.

> This is an excellent suggestion and we now clarify the speed and applicability of our current two-step recycling process (see p7, line 53, and p8, line 2). We also include a time axis for Figure 4e. Indeed, we agree that compatibility with on-line methods is a key limitation of reusability methods. Currently we reach timescales of seconds, faster than other *in-situ* recycling methods reported in the literature. At current speeds, our method is suited to off-line sequential detection of fractionated chromatographic eluates. However this is not a fundamental limit, and we are currently working to further minimise the regeneration time to improve its suitability for rapid on-line detection, now noted in the manuscript (p11, lines 25-26). The current method thus presents a significant advance towards quantitative, reproducible, sequential flow analysis using SERS.

3. Page 1, line 46: The claim that all the methods mentioned lead to morphological changes on the surface of the substrate is not substantiated.

> We thank the reviewer for pointing this out and have revised this line accordingly.

4. Page 2 line 8: The word "scaffolding" is still unclear in the text and should be defined earlier to simplify the understanding of the text.

> We have now revised the text to clarify the meaning of 'scaffolding' earlier (p2, line 9).

5. Page 2 line 24: The benefits mentioned should be defined in the text.

> We now clarify the advantages of thin-film aggregates for in-flow SERS sensing (p2, line 28-30). Specifically, these thin-film aggregates can be readily integrated into flow cell systems and can be probed optically from the backside when deposited on optically transparent solid supports.

6. The experimental setup shown in Figure S3 should be included in the main article.

> We thank for the reviewer for this suggestion and now include a diagram of our EC-SERS flow set-up in Figure 1.

7. Page 10: The standard addition procedure is explained in great detail. All related paragraphs can be deleted as the standard addition procedure is common when dealing with samples containing complex matrices.

> This is helpful, and as suggested we cut down our discussion of the standard addition method.

8. Please include a picture of the optical setup used in the SI.

> We now include a schematic and picture of the optical setup in the SI, as shown below (Figure R1).

Figure R1. Electrochemical SERS (EC-SERS) flow and optical set-up. (a) Schematic diagram and photos of the (b) EC-SERS flow and (c) optical set-up.

9. Page 12: Analytical detection: Please indicate the conductivity of the buffer used.

> We now revise the manuscript to include the conductivity of the buffer (0.5 mS/cm) used.

10. S5 Graph b should be shown as a spectroelectrochemistry graph. The Raman intensity of each wavelength should be shown in relation to the applied potential. An example would be Figure 2 in the following publication: [10.1016/j.jelechem.2020.114743](https://doi.org/10.1016/j.jelechem.2020.114743)

> This is an extremely helpful suggestion. We now revise Figure S5 to include a SERS spectro-voltammogram. We also include plots of SERS spectra at specific applied potentials to more clearly highlight spectral features discussed in the text.

11. Page 6: At the end of the first paragraph it is mentioned that CB5 is desorbed in favour of anions

in the EDL when a positive potential is applied. Is it possible that CB5 undergoes an electrochemical reaction and is reduced before desorption?

> This is a valuable question. Prior work on CB[5] as well as considerations of its well known chemistry¹ suggest that it is extremely unlikely for CB[5] to undergo any electrochemical reaction at the range of applied potentials we investigate. Previous work reporting the use of CB[5] as an electrode modifier in aqueous electrolyte also shows that CB[5] is remarkably robust even at oxidising potentials inducing the oxygen evolution reaction^{2,3}. However, ions are known to coordinate at the CB[5] carbonyl portals, and these may change with electrochemical potential. We thus revise the manuscript to mention this possibility.

12. Page 8: Please add at which wavenumbers the phosphate ion peaks are visible.

> As suggested we now label wavenumbers of the phosphate ion peaks (1100 cm⁻¹). We also include plots of relevant SERS spectra which more clearly highlight the phosphate ion peaks.

13. Page 11: I recommend plotting this diagram as a spectrovoltammogram, as previously described

> As before, we are happy to adopt this suggestion, and now include a plot of the SERS spectro-voltammogram of 10 μM adenine (ADN) in buffer.

Reviewer #2 and #3 combined:

1. The electrochemical method has been used to regenerate the SERS substrate. Please add more discussions about the differences between the previous work and highlight the advantages of this work. I suggest giving more discussion about the mechanism of the action of cucurbit [5] uril (CB [5]) ligands in the work.

> This is an excellent suggestion, and we now include further discussions highlighting the differences and advantages of this work compared to our previous work (see p2, line 39, 45-47, 49-51) well as the work of others (see p4, line 51-53; p10, lines 48-51).

The method is based on our previous approach of using oxygen plasma cleaning to decompose adsorbates, followed by a re-scaffolding step in which we reintroduce a nanogap stabilising ligand.⁴ While this strategy is effective for cleaning and defining the surface functionalisation of the MLAGG SERS substrate, the process requires long treatment times and is incompatible with continuous measurements. Here, we adapt this into an *in-situ* electrochemical approach, which is 200x faster than previously and compatible with continuous flow systems. The improved practicality of this approach allows us to demonstrate rapid, effective, and reproducible recycling of the MLAGG-CB[5] SERS substrate.

Compared to other previously reported recycling methods, we demonstrate rapid (<30s) removal of a broad range of analytes (including thiols) using electrochemical oxidation, and the reproducible regeneration of the SERS hotspots using a molecular scaffold, cucurbit[5]uril. The level of both cleaning efficiency and reproducibility demonstrated in our approach has been inaccessible using other methods.

We also include additional discussion on how the CB[5] ligands act in our regeneration process (see p7, lines 6-9). Specifically, we propose that the high Au binding affinity of CB[5] ligands ensures that CB[5] ligands immediately bind to and stabilise the bare Au after Au oxide reduction. The rigid 0.90 nm height of the CB[5] molecule and its favourable binding in-between the two AuNP facets⁵ enables reproducible reconstruction of the nanogap hotspots after oxidative cleaning. The application of such a molecular scaffold to reproducibly reconstruct the nanogap hotspots after oxidation has not been reported before and is the key novelty of our approach.

2. In this work, the authors state that the gold can be oxidized during the electrochemical redox reactions. Will the morphology change after the 30 times cycles? It would be good to characterize the nanostructures before and after the cycles.

> A key step in our proposed SERS substrate cleaning and regeneration method is the application of an oxidising potential which desorbs and/or oxidises adsorbates from the SERS substrate. This also oxidises the surface of the AuNPs, thus a second step reduces the Au oxide and restabilises the nanogap hotspots with the CB[5] ligand. Indeed, since Au oxidation and reduction is known to induce surface morphology changes^{6,7}, we agree it is important to characterise the nanostructures before and after cycling. In Figure 3, we present SEM images and dark field scattering spectra to characterise the nanostructure morphology before and after cycling, which show that the nanogap hotspots and plasmonic coupling appear to remain consistent throughout. This contrasts with control experiments where Au is oxidised and subsequently reduced *without* the presence of CB[5]: here progressive sintering of the nanogaps is seen in SEM images, DF scattering spectra, and SERS. We note that SEM imaging shows some roughening of the outer surface of the AuNP with and without CB[5], but nanogap sintering is never seen with CB[5]. These results highlight the importance of reintroducing CB[5] to retain the nanogap hotspot morphology and therefore the SERS enhancement of the MLagg SERS substrate over multiple cycles.

3. It is well known that the thiols have a high affinity with gold. When exposed to thiols, is it possible that the thiol molecules may replace the CB in the nanovoids?

> Indeed this is an interesting question. In Figure 4c, we show SERS spectra of the MLagg after incubation in various 1 mM aromatic thiol solutions for 1 hour. In all cases, the CB[5] peak at 830 cm^{-1} decreases, suggesting that thiols can indeed replace some CB[5] molecules in the nanogap. However, as CB[5] peaks are still visible in SERS, this implies that not all CB[5] molecules are displaced. In **Figure R2**, we show that even after 24-hour incubation in 1 mM 1,1-biphenyl-4-thiol (BPT) solution in ethanol, the CB[5] peaks at 451 cm^{-1} and 830 cm^{-1} (highlighted yellow) are still evident. We believe the remaining CB[5] molecules are either (1) lodged deep within the nanogaps and kinetically constrained from thiol replacement, and/or (2) strongly bound to the nanogap due to binding carbonyl portals at either end with both AuNP facets. We thus briefly discuss this in the manuscript.

Figure R2. SERS spectra of MLagg-CB[5] before (black) and after (green) incubation in 1 mM 1,1-biphenyl-4-thiol (BPT) for 24 hrs. The highlighted peaks correspond to characteristic CB[5] peaks at 451 cm^{-1} and 830 cm^{-1} . Spectra are collected with 1 s integration time and 785 nm laser excitation with 1 mW power.

4. Considering the solid SERS substrate, what about the uniformity of the SERS measurements? Does the SERS substrate can keep uniformity after the cycling tests? To give the SERS mapping data would be more convincing.

> This is an extremely helpful comment, and we now include additional SERS mapping data at different points of the cycling. These show the local uniformity of the SERS substrate remains mostly unchanged after undergoing multiple cycles with CB[5] regeneration (**Figure R3**), giving an average analyte peak area between cycles of 14.6 ± 0.3 a.u. and local uniformity ranging from 6.2-7.8% RSD. This demonstrates that multiple hotspots across the SERS substrate are effectively regenerated. However, when no CB[5] is added during regeneration (**Figure R4**), we observe not only variation in the analyte signal intensity, but also a degradation in the uniformity of the SERS measurements from 6% RSD in cycle 1 to 12% in cycle 2, and to 29% RSD in cycle 10. We thus add these data to the SI.

Figure R3. Regional uniformity of ADN signal on MLagg-CB[5] over multiple cycles of analyte detection and cleaning/regeneration with CB[5]. (a) SERS spectrum of MLagg-CB[5] before analyte cycling. (b) Optical microscope image of 465x330 μm region-of-interest used for SERS mapping. For cycles 1 (c), 10 (d), 20 (e), and 30 (f): (i) SERS spectra of MLagg-CB[5] after 5 μM ADN binding at -0.60 V in 50 mM potassium phosphate buffer (pH 7.0) (red), and after cleaning and regeneration with CB[5] (black). Spectra taken in-situ with 1s integration time, 1mW 785 nm laser and 40x objective. Heatmaps of SERS ADN peak area ($\nu_{ADN} = 732$ cm⁻¹) over the dried MLagg-CB[5] surface area: (ii) after ADN, and (iii) after ReSERS with CB[5]. Heatmaps taken over region shown in (b, right) on a 31x11 grid with 15x30 μm spacings. SERS map captured using 1s integration time, 785 nm excitation laser, 2.1 mW power with a 20x objective.

Figure R4. Regional uniformity of ADN signal on MLagg-CB[5] over multiple cycles of analyte detection and cleaning/regeneration without CB[5]. (a) SERS spectrum of MLagg-CB[5] before analyte cycling. (b) Optical microscope image of 465x330 μm region-of-interest used for SERS mapping. For cycles 1 (c), 2 (d), and 10 (e): (i) SERS spectra of MLagg-CB[5] after 5 μM ADN binding at -0.60 V in 50 mM potassium phosphate buffer (pH 7.0) (red), and after cleaning and regeneration without CB[5] (black). Spectra are taken in-situ with 1s integration time, 1 mW 785 nm laser using a 40x objective. Heatmaps of SERS ADN peak area ($\nu_{ADN} = 732 \text{ cm}^{-1}$) over the dried MLagg-CB[5] surface area: (ii) after ADN binding, and (iii) after ReSERS without CB[5]. Heatmaps taken over 465x330 μm region shown in (b, right) on a 31x11 grid with 15x30 μm spacings. SERS map recorded using 1 s integration time, 785 nm excitation laser, 2.1 mW power with a 20x objective.

5. The article only mentions renewable data with 30 repetitions. Is it the maximum limit number of repetitions for this base?

> While we show 30 continuous analyte detection and cleaning/regeneration cycles in the manuscript, we note that we have successfully reused MLagg-CB[5] SERS substrates beyond 100 non-continuous cycles (i.e., with intermittent removal from the EC-SERS flow cell, followed by rinsing and drying). We briefly discuss possible limitations now in the manuscript (see p4, lines 52-53).

6. The relevant parameters of SERS measurements should be supplemented.

> We now include details of our SERS measurement parameters throughout the manuscript, as highlighted in the text.

7. The related work maybe help to strengthen the research background, such as, Recyclable Surface-Enhanced Raman Scattering Substrate-Based Sensors, ACS Sustainable Chemistry & Engineering, (2023), DOI: 10.1021/acssuschemeng.2c05291, and moreover, please modify the reference format carefully, i.e. the reference format needs to be unified according to the journal format requirements. > This is a relevant related work which indeed strengthens the research background and now included. We corrected the reference formatting in accordance with the journal requirements.

References

1. McCune, J. A., Rosta, E. & Scherman, O. A. Modulating the oxidation of cucurbit[n]urils. *Org. Biomol. Chem.* **15**, 998–1005 (2017).
2. Li, F. *et al.* A Cobalt@Cucurbit[5]uril Complex as a Highly Efficient Supramolecular Catalyst for Electrochemical and Photoelectrochemical Water Splitting. *Angew. Chemie* **133**, 2004–2013 (2021).
3. Zhao, Y. *et al.* Manipulating the Host–Guest Chemistry of Cucurbituril to Propel Highly Reversible Zinc Metal Anodes. *Small* **2308164**, 1–10 (2023).
4. Grys, D.-B. *et al.* Controlling Atomic-Scale Restructuring and Cleaning of Gold Nanogap Multilayers for Surface-Enhanced Raman Scattering Sensing. *ACS Sensors* **8**, 2879–2888 (2023).
5. Taylor, R. W. *et al.* Precise Subnanometer Plasmonic Junctions for SERS within Gold Nanoparticle Assemblies Using Cucurbit[n]uril “Glue”. *ACS Nano* **5**, 3878–3887 (2011).
6. Tran, T. D. *et al.* Restructuring a gold nanocatalyst by electrochemical treatment to recover its H₂ evolution catalytic activity. *Sustain. Energy Fuels* **5**, 1458–1465 (2021).
7. Wang, Y., Laborda, E., Crossley, A. & Compton, R. G. Surface oxidation of gold nanoparticles supported on a glassy carbon electrode in sulphuric acid medium: contrasts with the behaviour of ‘macro’ gold. *Phys. Chem. Chem. Phys.* **15**, 3133 (2013).

Reviewers' Comments:

Reviewer #1:

Remarks to the Author:

The present version is now fit for publication!

Reviewer #3:

Remarks to the Author:

I have thoroughly reviewed the manuscript, along with the author's responses to feedback from other reviewers. I am pleased to note that the manuscript has undergone substantial improvements, and the concerns I previously had have been effectively addressed in the revised draft. Given these improvements, I wholeheartedly recommend this article for publication in its current form.